# One versus two doses of ivermectin-based mass drug administration for the control of scabies: A cluster randomised non-inferiority trial

Susanna J. Lake[1,2,3]*, Daniel Engelman[1,2,3], Julie Zinihite[1], Oliver Sokana[4], Dickson Boara[4], Titus Nasi[4], Christina Gorae[4], Millicent H. Osti[1,2], Sophie Phelan[5], Matthew Parnaby[1], Anneke C. Grobler[6], Tibor Schuster[7], Ross Andrews[8], Margot J. Whitfeld[9], Michael Marks[10,11,12], Lucia Romani[1,5], Andrew C. Steer[1,2,3], John M. Kaldor[5]

1 Tropical Diseases Research Group, Murdoch Children's Research Institute, Melbourne, Australia, 2 Department of Paediatrics, University of Melbourne, Melbourne, Australia, 3 Melbourne Children's Global Health, Melbourne, Australia, 4 Ministry of Health and Medical Services, Solomon Islands, 5 Kirby Institute, University of New South Wales, Sydney, Australia, 6 Clinical Epidemiology and Biostatistics Unit, Murdoch Children's Research Institute, Melbourne, Australia, 7 Department of Family Medicine, McGill University, Montreal, Canada, 8 Australian National University, Canberra, Australia, 9 Department of Dermatology, St Vincent's Hospital, Sydney, Australia, 10 Clinic Research Department, Faculty of Infectious and Tropical Diseases, London School of Hygiene and Tropical Medicine, London, United Kingdom, 11 Hospital for Tropical Diseases, University College London Hospital, London, United Kingdom, 12 Division of Infection and Immunity, University College London, London, United Kingdom

* susanna.lake@mcri.edu.au

**Data Availability Statement:** All relevant data are within the paper and its Supporting Information files.

## Abstract

### Background

Mass drug administration (MDA) based on two doses of ivermectin, one week apart, substantially reduces prevalence of both scabies and impetigo. The Regimens of Ivermectin for Scabies Elimination (RISE) trial assessed whether one-dose ivermectin-based MDA would be as effective.

### Methods

RISE was a cluster-randomised trial in Solomon Islands. We assigned 20 villages in a 1:1 ratio to one- or two-dose ivermectin-based MDA. We planned to test whether the impact of one dose on scabies prevalence at 12 and 24 months was non-inferior to two, at a 5% non-inferiority margin.

### Results

We deferred endpoint assessment to 21 months due to COVID-19. We enrolled 5239 participants in 20 villages at baseline and 3369 at 21 months from an estimated population of 5500. At baseline scabies prevalence was similar in the two arms (one-dose 17·2%; two-dose 13·2%). At 21 months, there was no reduction in scabies prevalence (one-dose 18·7%; two-dose 13·4%), and the confidence interval around the difference included values substantially greater than 5%. There was however a reduction in prevalence among those

**Funding:** The RISE trial is funded by the National Health and Medical Research Council of Australia GNT1127297. DE, LR, JMK and ACS are supported by fellowships from the National Health and Medical Research Council of Australia. The funders had no role in study design, data collection and analysis, decision to publish, or preparation of the manuscript.

**Competing interests:** The authors have declared that no competing interests exist.

who had been present at the baseline assessment (one-dose 15·9%; two-dose 10·8%). Additionally, we found a reduction in both scabies severity and impetigo prevalence in both arms, to a similar degree.

## Conclusions

There was no indication of an overall decline in scabies prevalence in either arm. The reduction in scabies prevalence in those present at baseline suggests that the unexpectedly high influx of people into the trial villages, likely related to the COVID-19 pandemic, may have compromised the effectiveness of the MDA. Despite the lack of effect there are important lessons to be learnt from this trial about conducting MDA for scabies in high prevalence settings.

## Trial registration

Registered with Australian New Zealand Clinical Trials Registry ACTRN12618001086257.

## Author summary

We conducted a trial of one versus two doses of ivermectin based mass drug administration (MDA) for the control of scabies. We found that no reduction in scabies prevalence in either the one or two dose arms. However there was a reduction in the severity of scabies and prevalence of impetigo in both arms, to a similar degree.

## Introduction

Scabies, recognised by the World Health Organization (WHO) as a neglected tropical disease (NTD) since 2017, has an estimated annual global incidence of 455 million cases [1]. Higher levels of endemic scabies largely occur in low and middle-income countries of the tropics, with Pacific island countries particularly affected [2]. The disease is caused by infestation with the mite *Sarcoptes scabiei* var. *hominis*, which burrows into the skin causing skin lesions and intense itch. The resulting breaches in the skin barrier can lead to secondary bacterial infections of skin and soft tissue structures, most commonly as impetigo. Serious immune-mediated complications from *Streptococcus pyogenes* impetigo can occur, including acute glomerulonephritis and possibly acute rheumatic fever [3].

The public health strategy of mass drug administration (MDA), which involves treating the whole population for a disease, regardless of symptoms or other diagnostic indicators, has been widely used for the control and elimination of a number of globally important NTDs [4]. A growing body of evidence supporting the use of MDA for scabies in high prevalence settings has led to interim guidance that it be adopted where the population prevalence of scabies is over 10% [5,6]. This guidance is based on the use of the oral antiparasitic agent ivermectin [7,8]. Clinical guidelines for the treatment of individual patients with scabies recommend a second dose of ivermectin, 7–14 days after the first, to ensure killing of unhatched mites that are protected from the drug at the first dose [9,10]. At the community level, delivery of the second dose of ivermectin poses barriers to the implementation of MDA due to increased cost, complex logistics and potentially poorer adherence, particularly when populations are scattered in remote, hard-to-access villages. It is also more difficult to integrate a two-dose

regimen with MDA for other NTDs that are based on one-dose regimens [11]. There is some evidence that one-dose MDA with ivermectin for other NTDs may have led to a reduction in scabies burden [12–14]. A recent study conducted in Fiji compared one- and two-dose ivermectin in combination with albendazole and diethylcarbamizine as MDA with a screen and treat approach, as a secondary outcome [15]. The nested study found that all three strategies led to a reduction in scabies prevalence. We therefore designed the Regimens of Ivermectin for Scabies Elimination (RISE) trial to assess the effectiveness of one dose of ivermectin-based MDA compared to a two dose regimen [16].

## Methods

### Ethics statement

The trial had ethical approval from the Solomon Islands Health Ethics Review Board (HRE005/18) and Royal Children's Hospital Human Research Ethics Committee, Melbourne, Australia (38099A), and was prospectively registered with Australian New Zealand Clinical Trials Registry ACTRN12618001086257. We obtained written informed consent from all participants aged 18 years and above, and from parents/guardians of those aged less than 18 years.

We conducted this study in Western Province in Solomon Islands, a Pacific island country with an estimated population in 2021 of around 700,000 people [17]. The study was conducted in partnership with the Western Provincial Health Services. We used an open-label, cluster-randomised, non-inferiority design in 20 villages, as described in the published protocol [16].

The cluster-randomised design has been used for this study as we aim to assess the impact of a public health intervention on a community, rather than individual participants. The cluster-randomised design aims to replicate real-world conditions which can include population movement in and out of villages, coverage gaps and other factors that may influence the outcome. The study will measure the change in scabies prevalence on a village level rather than tracking individuals.

Villages were randomly assigned by an independent statistician, in a 1:1 ratio, to MDA with either one dose (Arm 1) or two doses administered 7–14 days apart (Arm 2) of scabies treatment (Fig 1). Criteria for village selection included a population between 180–300 (based on census-derived estimates), physical separation from other villages, and willingness to participate in the study as determined by consultation involving the Western Provincial Health Services and village leadership. All residents were eligible to take part in the trial. MDA was conducted from 13 May to 3 July 2019. Prevalence surveys for scabies and impetigo were conducted immediately prior to MDA (baseline) and the protocol specified repeat prevalence surveys at 12 months (primary outcome) and 24 months after MDA.

MDA for scabies used directly-observed oral ivermectin in 6 mg tablets, unless there were contraindications to ivermectin, in which case topical permethrin was used. Ivermectin was dosed within a range of 150 to 250μg/kg. Height-based dosing was used for children at least 90cm tall up to the age of 14 years, ranging from 3mg to a maximum of 12mg. Intervals for height based dosing were based on previously collected Solomon Islands height and weight data (90–112 cm: 3mg; 112-138cm: 6mg; 138-156cm: 9mg; >156cm:12mg). A standard dose of 12mg was used for those aged 15 years and over, but reduced to 9mg for those judged on visual assessment to be substantially underweight relative to height, and increased to 15mg for those judged to be substantially overweight [18]. Permethrin 5% cream was offered along with administration instructions to participants with the following contraindications to ivermectin: children aged less than 2 years or less than 90cm in height, pregnancy, breastfeeding a child under one week of age, treatment with warfarin, or severe illness. The dosing regimen for

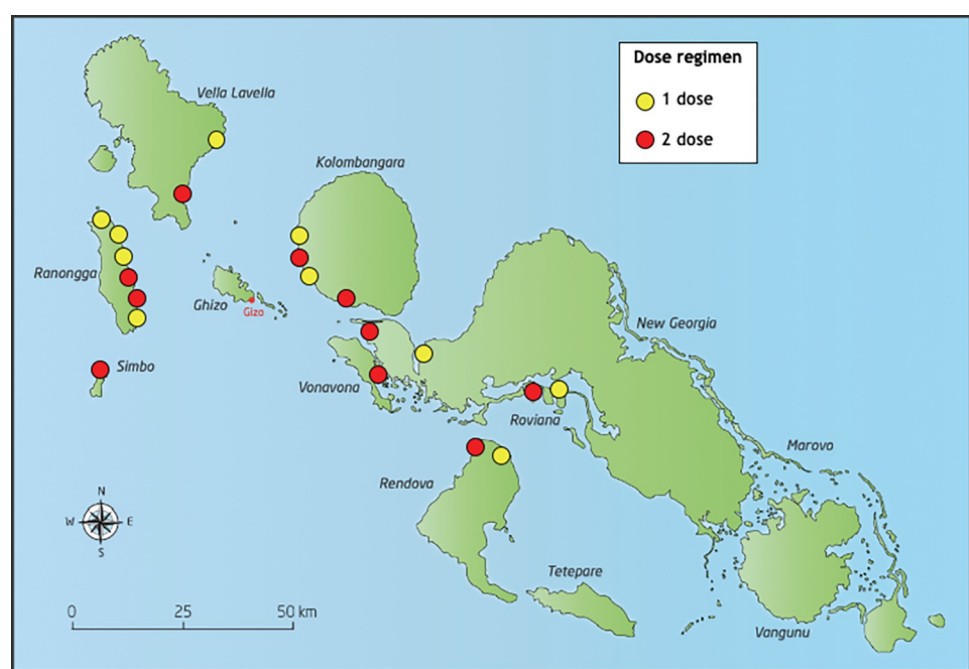

**Fig 1. Map of study villages in Western Province, Solomon Islands. Yellow dots identify villages in Arm 1 (1 dose) and red dots identify villages in Arm 2 (2 doses).** (The map in this figure was drawn by Hilary Bruce at Murdoch Children's Research Institute and adapted by the authors of the manuscript. The source that was used to create the map is freevectormaps.com, https://freevectormaps.com/solomon-islands/SB-EPS-02-0001. The authors understand and agree to the terms of the Creative Commons Attribution License).

permethrin was the same as that for ivermectin and determined by the village allocation; either one dose (Arm 1) or two doses 7–14 days apart (Arm 2).

Restrictions on travel and research related to the COVID-19 pandemic made it impossible to complete the 12-month survey. Therefore, we combined the proposed 12 and 24 month assessments into a single assessment at 21 months, which took place from 19 February to 21 April 2021. At both baseline and 21 months, clinical assessments for symptoms and signs of scabies were conducted by nurses who had received specific training on standardised guidelines [19]. Nurses examined the arms and legs of all participants, and also the trunk in children aged less than 2 years. Based on the clinical assessments, diagnoses were made against diagnostic criteria promulgated by the International Alliance for the Control of Scabies [20], as either clinical scabies (subcategory B3 in criteria) or suspected scabies (subcategories C1 and C2). Impetigo was defined as papular, pustular or ulcerative lesions surrounded by erythema, or with crusts, pus or bullae [21]. We determined the severity of scabies and impetigo by the number of lesions (very mild 1–2 lesions, mild 3–10 lesions, moderate 11–50 lesions, severe >50 lesions). At the 21 month survey participants were also asked whether they had participated in MDA at baseline. Individual data were entered directly into a REDCap database using Android-enabled mobile devices in the field, and securely stored at the Murdoch Children's Research Institute [22].

We also obtained routinely-collected data from the national District Health Information Software 2 (DHIS2), an electronic health management information system, on presentations for scabies and total presentation for all conditions, at all Western Province health clinics for the 12 months before and 12 months after MDA. The 12 month sampling window was chosen to assess the primary outcome at 12 months as per the original study design as well as to assess

any seasonal variation in scabies prevalence. Clinics were classified according to whether or not they served at least one study village. Health clinics service a combination of both study villages and non-study villages so the data collected from DHIS2 was used as a secondary measure to obtain clinic level data as a secondary outcome, not village or individual level data.

Although ivermectin and permethrin are well tolerated with no safety concerns at the dose used for MDA, we undertook passive monitoring for adverse events. Participants were advised to report any adverse events to health clinics or directly to the study team and these were reported to the study coordinator. We also retrospectively reviewed mortality and stillbirth records as reported in the national DHIS2 health information system. An independent Data Safety Monitoring Board (DSMB) reviewed these data.

The sample size was calculated using a standard Monte Carlo simulation and based on the previously reported scabies prevalence in the province and the effect size in studies for ivermectin-based MDA for scabies, with a non-inferiority margin of 5% absolute difference in prevalence to achieve statistical power of 80% [23]. We selected a non-inferiority margin of 5% as we considered this to be relevant from a public health perspective. This was an intention-to-treat analysis.

For the primary endpoint, we calculated scabies prevalence at baseline and 21 months for the two arms of the trial overall and for demographic sub-groups. The difference in scabies prevalence between baseline and 21 months was calculated for each village. The means of these differences were calculated for each study arm, and then the difference between these means, and its 95%, two-sided confidence interval was calculated. If the upper limit of the 95% confidence interval of the two study groups comparing baseline to 21 months was less than or equal to 5% (the clinically relevant non-inferiority margin) one-dose regimen was considered non-inferior to two doses.

In further analyses, we calculated the difference in impetigo prevalence between baseline and 21 months for each study arm using the same method as we did for scabies. We compared the severity of scabies by calculating the proportion of participants with very mild, mild, moderate and severe disease and their two-sided 95% confidence interval of these at each time point and dose group. We compared the number of presentations for scabies at the health clinics with the total number of clinic presentations in the 12 months before and 12 months after MDA between clinics that serviced study villages and other health facilities in Western Province. Statistical analysis was conducted using Stata (version 17·0, StataCorp, College Station, TX, USA).

## Results

At baseline, the total estimated population of the 20 study villages was 5500 (Fig 1). The In Arm 1 villages, of an estimated population of 2600, 2418 participants enrolled at baseline and 2412 received one-dose MDA. In Arm 2 villages, estimated population 2900, baseline 2842 participants enrolled at baseline, 2837 received the first dose of MDA and 2346 received a dose at the second timepoint 7–14 days later (including 189 who had not received the first dose). At 21 months, 1699 individuals were assessed in Arm 1, including 804 who reported that they had been recruited at baseline and 877 (51·6%) who reported that they had not. In Arm 2, 1692 were assessed including 1057 recruited at baseline and 622 (36·8%) new (Fig 2). There was a higher proportion of females than males in each arm and at both time-points, notably there was an underrepresentation of adult males in the 20–49 year old age groups (Table 1). There was a higher proportion of children under the age of 15 years participating at 21 months compared to baseline.

There was no significant change in scabies prevalence in either arm at the 21-month survey, compared to baseline. In Arm 1, prevalence was 17·2% (95% CI 15·7–18·7) at baseline and

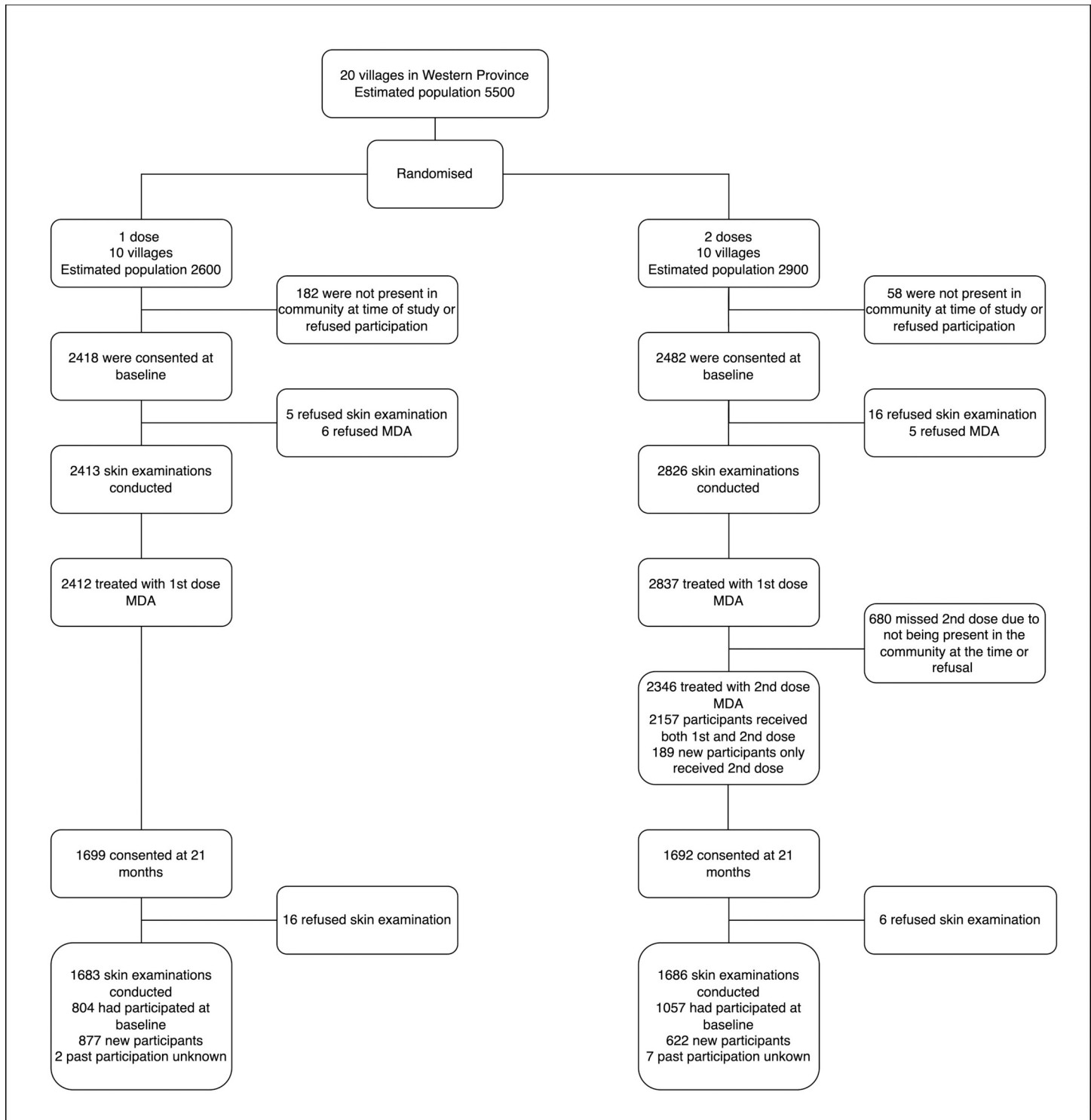

**Fig 2. CONSORT flow diagram detailing participation in study.** MDA: Mass Drug Administration.

18·7% (95% CI 16·8–20·6) at 21 months. In Arm 2, prevalence was 13·2% (95% CI 12·0–14·5) at baseline and 13·4% (95% CI 11·8–15·1) at 21 months. The difference between the two arms

**Table 1. Demographic characteristics of participants in the study at baseline and at 21 months.**

|  | Arm 1 (1 dose MDA) | | Arm 2 (2 dose MDA) | |
| --- | --- | --- | --- | --- |
|  | Baseline N (%) | 21 months N (%) | Baseline N (%) | 21 months N (%) |
| **Total** | 2413 | 1683 | 2826 | 1686 |
| **Participation at baseline (self-reported)** |  | 804 (47·8) |  | 1057 (62·9) |
| **Sex** |  |  |  |  |
| **Female** | 1271 (52·7) | 983 (58·4) | 1491 (52·8) | 974 (57·8) |
| **Male** | 1142 (47·3) | 700 (41·6) | 1335 (47·2) | 712 (42·2) |
| **Age (years)** |  |  |  |  |
| **0–1** | 109 (7·5) | 67 (4·0) | 154 (5·5) | 72 (4·3) |
| **2–4** | 238 (9·9) | 130 (7·7) | 304 (10·8) | 143 (8·5) |
| **5–9** | 399 (16·5) | 343 (20·4) | 505 (17·9) | 352 (20·9) |
| **10–14** | 347 (14·4) | 392 (23·3) | 459 (16·2) | 338 (20·0) |
| **15–19** | 172 (7·1) | 148 (8·8) | 211 (7·5) | 149 (8·8) |
| **20–29** | 273 (11·3) | 142 (8·4) | 336 (11·9) | 135 (8·0) |
| **30–39** | 281 (11·7) | 155 (9·2) | 327 (11·6) | 169 (10·0) |
| **40–49** | 216 (9·0) | 115 (6·8) | 230 (8·1) | 135 (8·0) |
| **50–59** | 182 (7·5) | 99 (5·9) | 160 (5·7) | 95 (5·6) |
| **60+** | 196 (8·1) | 92 (5·5) | 140 (5·0) | 98 (5·8) |

MDA–mass drug administration

in the mean changes in scabies prevalence from baseline to 21 months was 0·8% (95% CI -9·3–11·0).

There was a lower scabies prevalence in participants at 21 months who were recruited at baseline compared to those who had not (Fig 3). In Arm 1, the prevalence was 15·9% (95% CI 13·5–18·6) in participants who had been present at baseline and 21·2% (95% CI 18·5–24·1) in

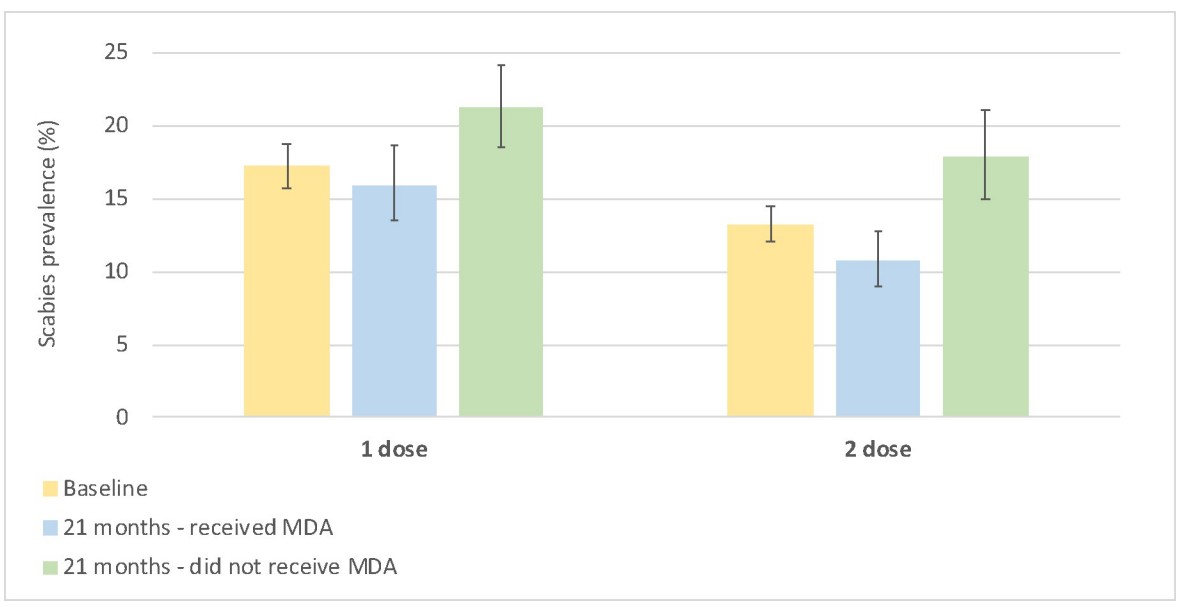

**Fig 3. Scabies prevalence before mass drug administration and at 21 months in participants who did and did not receive mass drug administration (error bars indicate 95% confidence interval).**

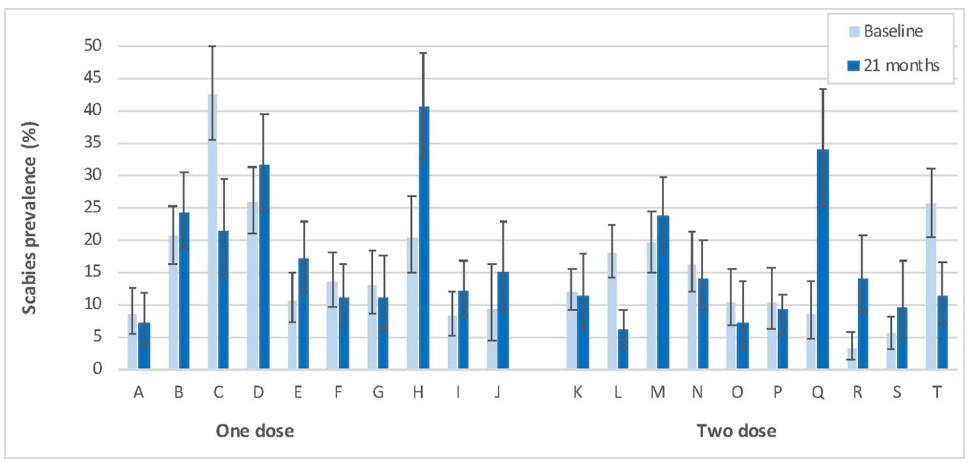

**Fig 4. Scabies prevalence before mass drug administration and at 21 months by villages (error bars indicate 95% confidence interval).**

those who had not. In Arm 2 the prevalence was 10·8% (95% CI 9·0–12·8) in participants who had been present at baseline and 17·8% (95% CI 14·9–21·1) in those who had not. The change in scabies prevalence from baseline to 21 months varied widely between villages (Fig 4), from an increase of 25·4% (village Q; baseline: 8·5% to 21 months: 33·9%), to a reduction of 12·1% (village L; baseline: 18% to 21 months 5·9%).

Scabies severity was reduced in both arms at 21 months compared to baseline (Table 2). In Arm 1, 20·1% of cases were classified as moderate-severe at 21 months compared to 41·7% at baseline. In Arm 2 19·1% of case were moderate-severe at 21 months compared to 57·1% at baseline.

The proportion of clinic presentations for scabies in clinics serving study villages was 0·41% for the 12-month period after MDA compared to 0·51% in the 12-month period before (Table 3). The absolute number of scabies presentations also fell from 242 to 162. In other clinics in the province the number and proportion of presentations for scabies were relatively unchanged.

**Table 2. Scabies and impetigo prevalence and severity at baseline and at 21 months after mass drug administration.**

|  | Arm 1 (One-dose MDA) | | Arm 2 (Two-dose MDA) | |
|---|---|---|---|---|
|  | Baseline N (%, 95% CI) | 21 months N (%, 95% CI) | Baseline N (%, 95% CI) | 21 months N (%, 95% CI) |
| **Scabies** | 414 (17·2, 15·7–18·7) | 314 (18·7, 16·8–20·6) | 373 (13·2, 12·0–14·5) | 225 (13·4, 11·8–15·1) |
| **Very mild (1–2 lesions)** | 20 (4·8, 3·0–7·4) | 5 (1·6, 0·5–3·7) | 17 (4·6, 2·7–7·2) | 3 (1·3, 0·3–3·8) |
| **Mild (3–10 lesions)** | 221 (53·4, 48·4–58·3) | 246 (78·3, 73·4–82·8) | 143 (38·3, 33·4–43·5) | 179 (79·6, 73·7–84·6) |
| **Moderate (11–50 lesions)** | 119 (28·7, 24·4–33·4) | 60 (19·1, 14·9–23·9) | 147 (39·4, 34·4–44·6) | 38 (16·9, 12·2–22·4) |
| **Severe (>50 lesions)** | 54 (13·0, 10·0–16·7) | 3 (1·0, 0·2–2·8) | 66 (17·7, 14·0–22·0) | 5 (2·2, 0·7–5·1) |
| **Impetigo** | 112 (4·6, 3·8–5·5) | 22 (1·3, 0·8–2·0) | 175 (6·2, 5·3–7·1) | 19 (1·1, 0·7–1·8) |
| **Very mild (1–2 lesions)** | 76 (67·9, 58·4–76·4) | 10 (40·9, 20·7–63·6) | 120 (68·6, 61·1–75·4) | 7 (36·8, 16·3–61·6) |
| **Mild (3–10 lesions)** | 27 (24·1, 16·5–33·1) | 11 (50·0, 28·2–71·8) | 52 (29·7, 23·1–37·1) | 12 (63·2, 38·4–83·7) |
| **Moderate (11–50 lesions)** | 6 (5·4, 2·0–11·3) | 2 (9·1, 1·1–29·2) | 3 (1·7, 0·4–4·9) | - |
| **Severe (>50 lesions)** | 3 (2·7, 0·6–7·6) | - | - | - |

MDA–mass drug administration

95% CI– 95% confidence interval

**Table 3. Presentations to Western Province health clinics in the year before and after mass drug administration (total presentations, presentations diagnosed as scabies and proportion of scabies presentations compared to total presentations).**

| | July-Sep. 2018 | Oct.-Dec. 2018 | Jan.-March 2019 | April-June 2019 | Total 12 months before MDA | July-Sep. 2019 | Oct.-Dec. 2019 | Jan.-March 2020 | April-June 2020 | Total 12 months after MDA |
|---|---|---|---|---|---|---|---|---|---|---|
| **Clinics servicing MDA villages** | | | | | | | | | | |
| Total presentations | 11,611 | 11,314 | 12,126 | 12,424 | **47,475** | 9383 | 9026 | 10,630 | 10,950 | **39,989** |
| Scabies presentations | 77 | 71 | 37 | 57 | **242** | 53 | 40 | 28 | 41 | **162** |
| Proportion of scabies presentations (%) | 0·66 | 0·63 | 0·31 | 0·46 | **0·51** | 0·56 | 0·44 | 0·26 | 0·38 | **0·41** |
| **Clinics not servicing MDA villages** | | | | | | | | | | |
| Total presentations | 27,312 | 28,042 | 28,968 | 29,029 | **113,351** | 24,685 | 25,146 | 28,693 | 28,915 | **107,439** |
| Scabies presentations | 170 | 73 | 123 | 109 | **475** | 174 | 112 | 114 | 109 | **509** |
| Proportion of scabies presentations (%) | 0·62 | 0·26 | 0·42 | 0·38 | **0·42** | 0·7 | 0·45 | 0·4 | 0·38 | **0·47** |

MDA–mass drug administration

Impetigo prevalence was significantly lower in both arms at 21 months after MDA compared to baseline (Table 2). In Arm 1, prevalence was 4·6% (95% CI 3·8–5·5) at baseline and 1·3% (95% CI 0·8–2·0) at 21 months. In Arm 2, prevalence was 6·2% (95% CI 5·3–7·1) at baseline and 1·1% (95% CI 0·7–1·8) at 21 months. The difference between the two arms in the mean changes in impetigo prevalence from baseline to 21 months was 1·7% (95% CI -2·8–6·2). The majority of cases were classified as very mild or mild and there was little change in impetigo severity in either arm from baseline to 21 months.

One adverse event was reported to the study team following MDA. The participant experienced nausea and vomiting which was self-resolving and did not require intervention.

## Discussion

In this trial of one versus two doses of ivermectin for scabies control, we did not observe a change in scabies prevalence between baseline and 21 months following MDA in either arm. The upper limit of the 95% confidence interval of the difference between the two study groups comparing baseline to 21 months was greater than 5% so we could not conclude that a single dose was non-inferior. However, there was a clear reduction in scabies severity in both arms, and clinics servicing study villages reported a modest decline in consultations for scabies in contrast to other clinics of the province which reported no change.

Our initial plan to conduct follow-up visits at both 12 and 24 months had to be modified when the Solomon Islands Ministry of Health and Medical Service paused health research in 2020 to minimise risks and maximise resources directed at the COVID-19 pandemic response. The 21-month procedures were unchanged from the planned for 12 and 24 months, apart from the remote involvement of the Australia-based study coordinator (SL).

The lack of reduction in scabies prevalence following ivermectin-based MDA in either arm was unexpected. In trials conducted in other provinces in Solomon Islands, we have recorded substantial declines attributed to this intervention and similar findings have been reported from studies in Fiji [7,8,24,25]. While it is possible that the results of these past studies do not accurately reflect the impact of MDA on scabies prevalence, it is more likely that other factors reduced the apparent impact of ivermectin-based MDA in our study.

Other studies in the region have shown significant reduction in scabies prevalence 24 and 36 months after MDA so it is unlikely that the follow up period was too long to capture the impact of MDA [26,27]. While we did not observe a decline in scabies prevalence at 21 months, there were reductions in both severity of scabies and prevalence of impetigo. One possible explanation is that there was an initial reduction in both scabies and impetigo prevalence, but by the time the delayed follow up assessment took place at 21 months, new cases of scabies had been introduced. These new cases introduced to the participating communities may have been at a milder clinical stage of scabies infestation, thereby providing an explanation for both the reduced scabies severity and the reduced impetigo prevalence. There was a reduction in scabies prevalence among the subset of participants who reported they had been present at baseline, which also suggests that scabies cases may have been introduced into the study villages after MDA.

Lower numbers of participants at 21 months, combined with a high proportion of participants in both arms reporting they had not been present during the MDA, indicate substantial movement both in and out of villages. As the study was designed to assess change in scabies prevalence at the village level, rather than the individual level, we were unable to link participant data between baseline and 21 months so relied on participants self-reporting if they had participated at baseline. There is evidence that people can accurately recall whether or not they received MDA up to 12 months after treatment, there is not published information on recall accuracy at longer time intervals [28]. Past studies in similar settings that have shown substantial reductions in scabies prevalence after MDA have treated whole islands, or additionally provided treatment to any new entrants to the community [7,8,25,29]. An earlier Solomon Islands study found a significant reduction in scabies up to 36 months after MDA treatment was offered to all province residents, reducing the likelihood of transmission through interaction with neighbouring villages in the province [8]. In 2020, as a response to the COVID-19 pandemic, the Solomon Islands Government requested that people return to their villages from urban areas, anecdotally resulting in a substantial influx of people to several study villages [30]. It is unclear what the results would have been if this trial had not coincided with the COVID-19 pandemic. In our trial, some village residents regularly attended markets in neighbouring villages to sell produce, and teenage children travelled away to boarding school, returning to the village for holidays. Logging camps established near several study villages between baseline and 21 months employed village residents. Movement of people in and out of study villages may have re-introduced scabies to these communities [31]. An important lesson from this trial may be that in settings with very high prevalence of scabies, MDA needs to go beyond individual villages, to encompass whole populations that have interlinked connections via school, work and other forms of engagement.

The ivermectin and permethrin that we used for this study were from international suppliers and met appropriate international quality standards including Good Manufacturing Practice (GMP). The study team directly observed participants taking ivermectin. Therefore, it is unlikely that either the quality of the treatments or adherence had any impact on the study outcomes.

There are a small number of health clinics that service the area of the study villages. We were unable to differentiate presentations by study arm as each clinic services multiple villages and aggregate consultation data at the clinic level. The small reduction in the proportion of scabies presentations in the 12 months after MDA in the clinics that service study villages compared to those that do not may further support the hypothesis that there was an initial decline in scabies prevalence following MDA, but it was undetectable at 21-months due to the unexpected population movement and the presumed re-introduction and transmission of scabies.

While the timing of trial procedures was affected by the COVID-19 pandemic, our procedures remained essentially intact. Beyond our control was the increased population movement which appears to have reduced the effectiveness of the ivermectin-based MDA strategy that has previously worked so well in Solomon Islands and other Pacific Island countries. Despite the challenges of this trial, recent results from a trial in Fiji indicate that single dose ivermectin is highly promising as a more practical means of delivering MDA [15]. Despite the exceptional circumstances faced by the RISE trial, our observations can serve to inform the design of scabies control programs and trials of MDA strategies.

## Supporting information

**S1 Table. Scabies prevalence at 21 months by demographic group according to self-reported participation in mass drug administration.**
(DOCX)

**S2 Table. CONSORT 2010 checklist of information to include when reporting a cluster randomised trial.**
(DOCX)

## Acknowledgments

The authors are grateful to the 20 communities in Western Province, Solomon Islands for their participation in this study. We appreciate the support and contributions of the Solomon Islands MHMS and Western Province Health Service including Pauline McNeil, Nemia Bainivalu, Gregory Jilini, Michael Larui, Ivan Ghemu, Freda Pitakaka, William Horoto, Jeffrey Korini, Soraya Pina, Yvonne Tuni, Selina Maena and Frederick Neqo. The RISE study team includes Sana Bisili, Aisling Byrne, Sharmillah Jack, Arthur Keremama, Alam Khatak, Erica Lazu, Brandon Le, Relinta Manaka, Davis Pesala, Deanne Seppy, Winter Sino, Patson Solomon, Ayleen Sosopu, Stephen Tiazi and Salote Wickham. This study is dedicated to Dr Tenneth Dalipanda, former Permanent Secretary of the Solomon Islands MHMS. Dr Tenneth was committed to improving the health of all Solomon Islanders. He was an advocate for public health research and his support enabled us to conduct this research and many studies before it.

## Author Contributions

**Conceptualization:** Daniel Engelman, Oliver Sokana, Titus Nasi, Anneke C. Grobler, Tibor Schuster, Ross Andrews, Margot J. Whitfeld, Michael Marks, Lucia Romani, Andrew C. Steer, John M. Kaldor.

**Data curation:** Susanna J. Lake.

**Formal analysis:** Susanna J. Lake, Anneke C. Grobler.

**Funding acquisition:** Andrew C. Steer.

**Project administration:** Susanna J. Lake, Matthew Parnaby.

**Writing – original draft:** Susanna J. Lake.

**Writing – review & editing:** Susanna J. Lake, Daniel Engelman, Julie Zinihite, Oliver Sokana, Dickson Boara, Titus Nasi, Christina Gorae, Millicent H. Osti, Sophie Phelan, Matthew Parnaby, Anneke C. Grobler, Tibor Schuster, Ross Andrews, Margot J. Whitfeld, Michael Marks, Lucia Romani, Andrew C. Steer, John M. Kaldor.

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
