## [Decision Letter · Decision Letter 0]

12 Jan 2023

Dear Ms Lake,

Thank you very much for submitting your manuscript "One versus two doses of ivermectin-based mass drug administration for the control of scabies: A cluster randomised non-inferiority trial" for consideration at PLOS Neglected Tropical Diseases. As with all papers reviewed by the journal, your manuscript was reviewed by members of the editorial board and by several independent reviewers. The reviewers appreciated the attention to an important topic. Based on the reviews, we are likely to accept this manuscript for publication, providing that you modify the manuscript according to the review recommendations. 

Thank you for submitting this cluster-randomised non-inferiority trial of one versus two doses of ivermectin mass drug administration (MDA) for the control of scabies in Solomon Islands. As a result of the ensuing Covid-19 pandemic, there was a deviation from protocol and a single endpoint assessment was made at 21 months. In contrast to previous studies of MDA with ivermectin, no reduction in prevalence of scabies was found in either study arm. A plausible explanation for this is that there was an influx of new people into the villages during the study period as a result of the pandemic, especially as a reduction in prevalence was found among those who had been present at the baseline assessment. In addition a reduction in the severity of scabies and in the prevalence of impetigo was identified.

This paper is of interest to a wide readership as scabies is now a WHO-recognized neglected tropical disease and there is improved awareness of its significant disease burden globally, including its association with secondary bacterial infections and immune-mediated complications such as acute glomerulonephritis and acute rheumatic fever. The study is also of interest to a wide readership as it highlights some of the challenges of instituting MDA treatment for scabies under "real world" conditions.

Before the paper can be considered for publication, the authors should response to each of the Reviewers comments, especially Reviewer 1's comment that in both in the abstract and conclusions the authors should make the exceptional circumstances clearer together with lessons that can be learned. 

Additional minor comment: Line 362. needs rewording

Sincerely,

Michele Murdoch

Guest Editor

Joseph Vinetz

Section Editor

Thank you for submitting this cluster-randomised non-inferiority trial of one versus two doses of ivermectin mass drug administration (MDA) for the control of scabies in Solomon Islands. As a result of the ensuing Covid-19 pandemic, there was a deviation from protocol and a single endpoint assessment was made at 21 months. In contrast to previous studies of MDA with ivermectin, no reduction in prevalence of scabies was found in either study arm. A plausible explanation for this is that there was an influx of new people into the villages during the study period as a result of the pandemic, especially as a reduction in prevalence was found among those who had been present at the baseline assessment. In addition a reduction in the severity of scabies and in the prevalence of impetigo was identified.

This paper is of interest to a wide readership as scabies is now a WHO-recognized neglected tropical disease and there is improved awareness of its significant disease burden globally, including its association with secondary bacterial infections and immune-mediated complications such as acute glomerulonephritis and acute rheumatic fever. The study is also of interest to a wide readership as it highlights some of the challenges of instituting MDA treatment for scabies under "real world" conditions.

Before the paper can be considered for publication, the authors should response to each of the Reviewers comments, especially Reviewer 1's comment that in both in the abstract and conclusions the authors should make the exceptional circumstances clearer together with lessons that can be learned. 

Additional minor comment: Line 362. needs rewording

Reviewer's Responses to Questions

**Key Review Criteria Required for Acceptance?**

**Methods**

-Are the objectives of the study clearly articulated with a clear testable hypothesis stated?

-Is the study design appropriate to address the stated objectives?

-Is the population clearly described and appropriate for the hypothesis being tested?

-Is the sample size sufficient to ensure adequate power to address the hypothesis being tested?

-Were correct statistical analysis used to support conclusions?

-Are there concerns about ethical or regulatory requirements being met?

Reviewer #1: The essential difficulty with this study – which was well planned and conducted - is that both arms of the ivermectin dosing ie one versus two doses failed to produce an effect and therefore cannot be compared. In the case of the two dose regimen, which is that used in previous studies the drug did not produce the anticipated effect in reduction of scabies levels. The study single dose produced similar effects. The authors argue that large movements of populations during the period of the study accounted for the difference. This seems a likely explanation although other possibilities should be considered such as failure of community compliance or a high level of local ivermectin resistance in the Sarcoptes population – both are unlikely. This leaves a problem in that the study failed to achieve its objective and the purpose of publishing is to record reasons for study failure. The study for understandable reasons also failed to follow the study protocol leaving a single evaluation point at 21 months. 

In writing up the study both in the abstract and conclusions the authors should make the exceptional circumstances clearer together with lessons that can be learned. Reintroduction of scabies into mainland treated communities by untreated patients has already been highlighted as a public health challenge. This study should have provided the opportunity to emphasise this. Otherwise it simply reads as a well conducted but failed experiment. Some rewording is required

Reviewer #2: It is regrettably unusual for researchers to submit so called "negative studies", I congratulate this team on putting this together and persevering inspite of the challenges the pandemic brought.

The study design is clear as are the objectives, population etch. All ethical approval appears to have been sort from host and other countries.

Local/host country researchers are included in the study (not as first or senior author).

Reviewer #3: This is a well-written paper with very clear objectives. Though scabies was the main focus disease for the study the opportunities to study impetigo using this same method which was also nested to simulate real population dynamics were undertaken. This brings the results close to the reality on the ground. 

This research paper is also premised on the fact that ivermectin is impactful in the mass treatment of scabies, and for this reason, the WHO has started a program to address scabies elimination. However, this paper set out to explore a more efficacious treatment regimen of twice-a-year treatment as against once-a-year treatment. This in my opinion is novel and opens up for further research, and additionally informs guidelines development for scabies elimination in the field of NTDs. 

The study justification is clear with clear objectives to guide the method. The study methodology is adequate and justifies the results obtained despite the challenges associated with the COVID-19 pandemic that led to changes in the timelines and study methodology by deferring the endpoint. Despite the changes made enrollment into the study received adequate sample sizes that produced the results obtained with clear, logical, and convincing discussions. The statistical analyses are simple, easy to follow and understand and yet adequate.

**Results**

-Does the analysis presented match the analysis plan?

-Are the results clearly and completely presented?

-Are the figures (Tables, Images) of sufficient quality for clarity?

Reviewer #1: Please provide a reference for the case definition of impetigo used .

Can you provide details of the permethrin treatment regimen followed ?

Can you comment on the changes in impetigo data – do they provide any support for the view that population movements accounted for the unexpected results recorded.

Reviewer #2: The analysis presented deviates from the initial plan for reasons detailed in the methodology - the pandemic interupted the second phase of data collection so both post MDA data collections were amalagmated. This seems a reasonable work aroundl.

Reviewer #3: The analysis was changed to suit the altered methodology, but however, is very acceptable with clear results that were well presented very simple tables of sufficient quality and clarity.

**Conclusions**

-Are the conclusions supported by the data presented?

-Are the limitations of analysis clearly described?

-Do the authors discuss how these data can be helpful to advance our understanding of the topic under study?

-Is public health relevance addressed?

Reviewer #1: In other studies how significant has been failure to recall receipt of MDA treatment by patients as a factor in producing discordant results ?

What lessons does this work this provide for future studies or strategies for scabies control teams ?

Reviewer #2: The conclusions are supported by the data presented. The limitations of the analysis clearly described.

Reviewer #3: Despite the disruption caused by covid-19 pandemic, the outcomes are valuable to inform further research and decision-making. The research did adhere to all ethical review processes and standards of operation in my opinion.

The conclusions based on the results are still valuable in spite of the changes in methodology, and well presented in the light of the changing population dynamics having compromised the results but do open up further suitable research questions to be answered in other studies in similar settings and other geographies

**Editorial and Data Presentation Modifications?**

Reviewer #1: (No Response)

Reviewer #2: Page 13 line 264 - do you really need both table 2 and figure 3? I think there might be too many tables so suggest considering leaving out 2

Page 16 line 300 you mention impetigo is reduced post MDA etc referencing table 2 - I think you mean table 3

Reviewer #3: I encountered very few typos requiring editorial work in the paper.

**Summary and General Comments**

Reviewer #1: (No Response)

Reviewer #2: This RISE studyhas dealt with the challenges the pandemic brought to the execution of the data collection in a constructive way. The work introduces the first experience of an MDA for scabies apparently having little impact on overall scabies levels in the community inlike previous MDA studies - although these were more isolated communities/isolated and did not have the returnees joining midway through sent from the big cities back to thier home villages - which has been effecively explained as an interpretation.

Reviewer #3: The paper is well-written and does not require any significant changes except minimal editorial work as already indicated.

PLOS authors have the option to publish the peer review history of their article (what does this mean?). If published, this will include your full peer review and any attached files.

Reviewer #1: No

Reviewer #2: Yes: Dr L Claire Fuller

Reviewer #3: Yes: Nana Kwadwo Biritwum

Figure Files:

Data Requirements:

Reproducibility:

References

---

## [Editor Report · Decision Letter 1]

1 Mar 2023

Dear Dr. Lake,

We are pleased to inform you that your manuscript 'One versus two doses of ivermectin-based mass drug administration for the control of scabies: A cluster randomised non-inferiority trial' has been provisionally accepted for publication in PLOS Neglected Tropical Diseases.

Best regards,

Michele Murdoch

Guest Editor

Joseph Vinetz

Section Editor

We thank the authors for presenting this revision of the manuscript which has satisfactorily addressed the previous reviewers' comments.

---

## [Editor Report · Acceptance letter]

10 Mar 2023

Dear Ms Lake,

We are delighted to inform you that your manuscript, "One versus two doses of ivermectin-based mass drug administration for the control of scabies: A cluster randomised non-inferiority trial," has been formally accepted for publication in PLOS Neglected Tropical Diseases.

Best regards,

Shaden Kamhawi

co-Editor-in-Chief

Paul Brindley

co-Editor-in-Chief
